# Object-Based Shadow Index via Illumination Intensity from High Resolution Satellite Images over Urban Areas

**DOI:** 10.3390/s20041077

**Published:** 2020-02-17

**Authors:** Haoyang Fu, Tingting Zhou, Chenglin Sun

**Affiliations:** Coherent Light and Atomic and Molecular Spectroscopy Laboratory, College of Physics, Jilin University, Changchun 130012, China; fuhy16@mails.jlu.edu.cn (H.F.); ttzhou18@mails.jlu.edu.cn (T.Z.)

**Keywords:** shadow index, multi-scale segmentation, object-based, illumination intensity, shadow intensity, urban remote sensing

## Abstract

For multi-spectral remote sensing imagery, accurate shadow extraction is of great significance for overcoming the information loss caused by high buildings and the solar incidence angle in urban remote sensing. However, diverse solar illumination conditions, similarities between shadows, and other dark land features bring uncertainties and deviations to shadow extraction processes and results. In this paper, we classify shadows as either strong or weak based on the ratio between ambient light intensity and direct light intensity, and use the fractal net evolution approach (FNEA), which is a multi-scale segmentation method based on spectral and shape heterogeneity, to reduce the interference of salt and pepper noise and relieve the error of misdiagnosing land covers with high reflectivity in shaded regions as unshaded ones. Subsequently, an object-based shadow index (OSI) is presented according to the illumination intensities of different reflectance features, as well as using the normalized difference water index (NDWI) and near infrared (NIR) band to highlight shadows and eliminate water body interference. The data from three high-spatial-resolution satellites—WorldView-2 (WV-2), WorldView-3 (WV-3), and GaoFen-2 (GF-2)—were used to test the methods and verify the robustness of the OSI. The results show that the OSI index performed well regarding both strong and weak shadows with the user accuracy and the producer accuracy both above 90%, while the four other existing indexes that were tested were not effective at diverse solar illumination conditions. In addition, all the disturbances from water body were excluded well when using the OSI, except for the GF-2 data in weak shadows.

## 1. Introduction

High-resolution, multi-spectral remote sensing images are widely utilized when observing activities on earth, especially in urban areas, where shadows are inevitably generated by high buildings, trees, bridges, or lofty towers. Although shadow can be considered as meaningful information in the evaluation of a building’s height and 3D reconstruction, it may complicate change detection, land cover classification, and other scenes [1]. Therefore, developing an accurate and robust shadow extraction method is significant for feature identification and information recovery of remote sensing imagery [2,3,4]. Although the shadow extraction of remote sensing images has been widely studied [5,6,7], the research of the methods for high-resolution remote sensing imagery over urban areas is not thorough yet. The previous methods mainly focused on the extraction of cloud shadows [2,8,9,10,11,12,13,14,15], which are more suitable for moderate-resolution images, including Landsat and Sentinel-2 imagery. The approaches for shadow detection consist of model-based methods, feature-based methods, shadow index methods, and machine learning methods [5,16], which are described in detail below.

Model-based methods require the prior knowledge of the sensor’s/camera’s location and the light direction to construct mathematical models, such as digital surface models (DSMs) [17]. The quality of the DSMs affects the shadow extraction results, as well as the method used in practical applications. Nakajima et al. [18] united the DSM data and the position information in an IKONOS image, and then simulated shadow with the sun angle and azimuth. Gustav et al. [19] utilized a straight-forward line-of-sight analysis on a DSM to get a rough shadow, then applied a support vector machine (SVM) to detect an accurate shadow. Nevertheless, the prior knowledge information, such as the sun location, is lacking for many remote sensing images, which means that these methods are not convenient in many scenes [20].

Feature-based methods detect shaded areas using their features, which could be spectrum, texture and semantic information [21,22], and have been widely used due to their simplicity and immediacy. These methods are commonly pixel-based or object-based. The former one is based on reliable samples or suitable thresholds via visual inspection [23]. The other provides an effective method for uniting spatial information, texture information, and semantic information in feature detection [24,25], but may complicate the algorithms. Instead of processing the image with individual pixels directly, the object-based method segments the original image into independent objects in which the pixels are addressed as the same land cover in the following processes [26] using different segmentation algorithms [27,28,29,30]. The object-based method has been proved effective for shadow extraction and removal in remote sensing imagery after considerable research. For most cases, unshaded objects contain the primary features that can be used for relighting shaded objects. Guo et al. [31] found out the most similar unshaded object for each shaded object according to distance, texture, and the brightness gradient. They were confidently thought to be the same material and the features of the unshaded object were used to compensate the lacking information of the shaded object. Luo et al. [17] applied a multi-scale segmentation method to generate primitive objects, and three spectral properties of shadows were fused based on the Dempster–Shafer (D–S) evidence theory to identify shaded regions. Feature-based methods have been widely used, but many methods require human intervention [32] or samples [33], which will affect the accuracy and speed of the shadow extraction for large-scale remote sensing images.

Shadow index methods have a faster calculating speed compared with previous methods, especially for large-scale images. The method directly depends on the spectral information of the image and it works well when the spectral information is abundant. For an RGB image, plentiful methods have been proposed that involve converting images into different invariant color spaces (e.g., HSB (Hue, Saturation, Brightness), HSV (Hue, Saturation, Value), and HIS: (Hue, Intensity, Saturation) in which the shadow features can be reinforced to form shadow-identified solutions [31,34,35,36,37]. For example, Arévalo [38] proposed a new color space called the C1C2C3 and combined it with Hue, Saturation and Value (HSV) to extract shadow. Chung et al. [39] presented a novel successive thresholding scheme (STS) improving Tsai’s [6] algorithm that makes the shadow detection more accurate. Unlike RGB images, multi-spectral images with rich spectral information have more advantages that can be used to design a more accurate shadow extraction index. The NIR channel [5] has been proved to be greatly sensitive to shadows. Because the radiance received by sensors from shadowed regions decreases from short to long wavelengths due to scattering, adding the NIR to the shadow index will improve the accuracy of shadow extraction. All the shadow extraction methods in References [21,40,41,42,43] attempted to include the NIR channel to improve the shadow extraction accuracy. However, shadow indexes might not always be efficient because the spectral characteristics of shadows are disparate under different illumination conditions.

Besides the traditional algorithm-based approaches above, machine learning as a new method for image processing has been utilized for shadow detection recently [10,44,45,46]. For example, generative adversarial networks (GANs) has been proved to be one of the most appropriate networks for shadow detection and removal [47,48]. The methods based on GANs consist of the generator G (which is trained to produce a realistic image) and the discriminator D (which is trained to distinguish shadow-free images from the images produced by G). Through stacked adversarial components, the shadow mask can finally be generated. However, machine learning methods have high requirements in terms of training data and hardware, even if efficient. Therefore, it is inappropriate for shadow extraction from multi-spectral remote sensing imagery on a large urban scale.

The purpose of this study was to propose a shadow extraction method for high-resolution, multi-spectral remote sensing images that would be suitable for different light intensities at a large scale. The WorldView-2 (WV-2), WorldView-3 (WV-3), and GaoFen-2 (GF-2) data were used for the experiments, and this method can be extended to other multi-spectral images with at least R, G, B, and NIR channels, including QuickBird, Sentinel-2A, IKONOS, GeoEye, SPOT, Gaofen-1, and Jilin-1 data. Based on the foregoing research, the existing methods fail to take illumination conditions into account, resulting to a weak robustness in different situations. In strong illumination irradiation (especially in summer), the contrast between shaded and unshaded regions tends to be sharp and the spectral characteristics of land covers in shadow areas are not obvious. However, in low illumination irradiation (such as in winter), the distinction between shaded and unshaded areas is smaller, and therefore the spectra of land covers in shaded regions can be reflected prominently. The method in this paper combines an object-oriented method and a shadow index method. It is the first important attempt to define the intensity of shadow under different illumination conditions and to propose indexes suitable for different shadow intensities.

We mainly analyzed the spectral characteristics of shadow from different satellite data under different illumination intensities and proposed an object-based shadow index (OSI), which was proved to have an accuracy of over 90%. In order to verify the effectiveness of OSI, we compared it with the following indexes: the morphological shadow index (MSI) proposed by Huang et al. [49] for building/shadow extraction from WorldView-2 imagery; the combinational shadow index (CSI) by Sun et al. [42] for building shadow extraction from Sentinel-2A MSI imagery with a medium resolution; the shadow detection index (SDI) by Shahi et al. for WorldView-2 imagery with Blue, NIR1, and NIR2 bands [43]; and the normalized saturation-value difference index (NSVDI) by Ma et al. [50] in the hue-saturation-value (HSV) color space.

The novelty and main contribution of this study includes: (1) a method proposed to define the intensity of shadow according to illumination intensity, (2) an object-based method applied to relieve the error of misdiagnosing high-reflectivity land covers in shaded regions as unshaded, and (3) the shadow indexes proposed suitable for different multi-spectral remote sensing imagery based on the intensity of shadows.

## 2. Materials and Methods

### 2.1. Data Description and Preprocessing

The experimental data for this study were from WV-2, WV-3, and GF-2 imagery. The WV-2 data contain one panchromatic band with a resolution of 0.46 m and eight multi-spectral bands with a resolution of 1.84 m. Except for the panchromatic band with a resolution of 0.31 m and eight multi-spectral bands with a resolution of 1.24 m, the WV-3 satellite added eight short-wavelength infrared bands (SWIR) with a resolution of 3.7 m. GF-2, launched in China on 19 August 2014, has one panchromatic band and four multi-spectral bands with the resolutions of 0.81 m and 3.24 m, respectively.

For this study, one image from the WV-2 satellite (WV-2 dataset) and one image from the WV-3 satellite (WV-3 dataset) and two images from the GF-2 satellite (GF-2 Sep. dataset and GF-2 Nov. dataset) were selected for shadow spectral analysis and the shadow extraction test. The details of the five experimental datasets are shown in Table 1 and Table 2.

The four images were selected with cloud content less than 5%, and the preprocessing was executed before applying, including: (1) radiation calibration—converting the digital number (DN) of the image into radiance or reflectance, (2) Fast Line-of-sight Atmospheric Analysis of Spectral Hypercubes (FLAASH) atmospheric correction [51]—removing the influence of clouds and aerosols in the image and obtaining the true reflectivity of the landcover, and (3) image fusion—fusing panchromatic image and multi-spectral image to obtain high-resolution image with multi-spectral information.

### 2.2. Shadow Intensity

The value of a pixel in an image is the product of the illumination intensity and the reflectance: $V_{i}=R_{i}I_{i}$, where $V_{i}$ is the value of the observed RGB image at pixel i. $R_{i}$ and $I_{i}$ are the reflectance and illumination intensity of pixel i, respectively. We assumed that the illumination intensity *I* consisted of ambient light intensity and direct light intensity: $I=I_{d}+I_{e}$, where $I_{d}$ is direct light and $I_{e}$ is ambient light. Thus, when the land cover is in shadow, it only has ambient light intensity such that $I=I_{e}$. Therefore, for any pixel i, the value can be expressed as:(1)$$V_{i}{}^{\mathrm{shadow-free}}=(I_{d}+I_{e})R_{i}\approx \frac{\sum_{j=1}^{N}U_{j}}{N}$$
(2)$$V_{i}{}^{\mathrm{shadow}}=I_{e}R_{i}\approx \frac{\sum_{k=1}^{M}S_{k}}{M}$$
where $V_{i}{}^{\mathrm{shadow-free}}$ and $V_{i}{}^{\mathrm{shadow}}$ are the pixel-wise value of the unshaded region and shaded region, respectively; $U_{j}$ is the *j*th unshaded pixel; and *N* is the total number of unshaded pixels. $S_{k}$ represents the *k*th shaded pixel, and *M* is the total number of shaded pixels.

According to Equations (1) and (2), the ratio $r_{}$ between the ambient illumination intensity and the direct illumination intensity in an RGB image can be expressed as:(3)$$r_{}=\frac{V_{i}^{\mathrm{shadow-free}}-V_{i}^{\mathrm{shadow}}}{V_{i}^{\mathrm{shadow}}}\approx \frac{1}{3}\cdot (\sum_{R,G,B}\left(\frac{\sum_{j=1}^{N}U_{j}-\sum_{k=1}^{M}S_{k}}{\sum_{k=1}^{M}S_{k}}\right)).$$

We assumed that the illumination ratio r is positively correlated with the intensity of the shadow. The larger *r* is, the larger the ratio between the unshaded region and the shaded region, and the more difficult for the land covers in shadow to reflect their spectral features. For simplicity, with choosing *r* = 4 as the threshold, which also gave the best results for our datasets, we defined strong shadow as being *r* ≥ 4 and weak shadow as being *r* < 4. For multi-band remote sensing imagery, in order to get *r*, we combined its RGB bands and selected a white land cover that existed in both shaded and unshaded regions to calculated r. The samples for the *r* calculation of the four datasets are shown in Figure 1, in which the green regions were the selected white pixels from unshaded areas and the red ones were the white pixels covered by shadow.

### 2.3. Segmentation Analysis

#### 2.3.1. Segmentation Based on the Fractal Net Evolution Approach

In the image segmentation step, the fractal net evolution approach (FNEA) [1] embedded in eCognition [2,3,4] was applied in this study. The FNEA is a bottom-up region-merging strategy used to form homogenous regions beginning with single pixel. Considering the spectral heterogeneity $h_{\mathrm{spectrum}}$ and shape heterogeneity $h_{\mathrm{shape}}$ synthetically, a minimum consistency rule is proposed to evaluate the similarity between regions. Then, by adopting the local mutual optimal criterion and the bottom-up merging strategy in the merging process, the multi-scale and multi-level image segmentation is realized. Combined with heterogeneity measurement criteria for region merge, a comprehensive heterogeneity measurement criterion f is as follows:(4)$$f=\omega_{\mathrm{shape}}\cdot h_{\mathrm{shape}}+(1-\omega_{\mathrm{shape}})\cdot h_{\mathrm{spectral}}$$
where $\omega_{\mathrm{shape}}$ is the weighting factor with the range (0–1), which is used to balance the impact of spectral information and shape information.

For multi-spectral remote sensing images, the spectral merged heterogeneity metrics $h_{\mathrm{spectral}}$ in Equation (4) can be identified using:(5)$$h_{\mathrm{spectral}}=\sum_{c}\omega_{c}(n_{\mathrm{Merge}}\cdot \sigma_{c}{}^{\mathrm{Merge}}-(n_{\mathrm{obj}1}\cdot \sigma_{c}{}^{\mathrm{obj}1}+n_{\mathrm{obj}2}\cdot \sigma_{c}{}^{\mathrm{obj}2}))$$
where c is the number of band; $\omega_{c}$ indicates the weights in channel; $n_{\mathrm{obj}1}$, $n_{\mathrm{obj}2}$, $\sigma_{c}{}^{\mathrm{obj}1}$, and $\sigma_{c}{}^{\mathrm{obj}2}$ are the region’s size and standard deviation of two adjacent regions before being merged; and $n_{\mathrm{Merge}}$ and $\sigma_{c}{}^{\mathrm{Merge}}$ are the region’s size and standard deviation of the merged area.

As for the measurement criteria of shape heterogeneity, $h_{\mathrm{shape}}$ comprises the smoothness $h_{\mathrm{smooth}}$ and the compactness $h_{\mathrm{compact}}$. The former is used to characterize the smoothness of the merged region, and the other is to ensure that the merged region is more compact. The formulae of $h_{\mathrm{shape}}$ can be expressed as:(6)$$h_{\mathrm{shape}}=\omega_{\mathrm{compact}}\cdot h_{\mathrm{compact}}+(1-\omega_{\mathrm{compact}})\cdot h_{\mathrm{smooth}},$$
where $\omega_{\mathrm{compact}}$ represents the weight of the compact in calculating the shape difference metric with the range (0–1). Both $h_{\mathrm{compact}}$ and $h_{\mathrm{smooth}}$ can be computed using:(7)$$h_{\mathrm{smooth}}=n_{\mathrm{Merge}}\cdot \frac{l_{\mathrm{Merge}}}{b_{\mathrm{Merge}}}-(n_{\mathrm{obj}1}\cdot \frac{l_{\mathrm{obj}1}}{b_{\mathrm{obj}1}}+n_{\mathrm{obj}2}\cdot \frac{l_{\mathrm{obj}2}}{b_{\mathrm{obj}2}}),$$
(8)$$h_{\mathrm{compact}}=n_{\mathrm{Merge}}\cdot \frac{l_{\mathrm{Merge}}}{\sqrt{n_{\mathrm{Merge}}}}-(n_{\mathrm{obj}1}\cdot \frac{l_{\mathrm{obj}1}}{\sqrt{n_{\mathrm{obj}1}}}+n_{\mathrm{obj}2}\cdot \frac{l_{\mathrm{obj}2}}{\sqrt{n_{\mathrm{obj}2}}}),$$
where, for two adjacent regions, $n_{\mathrm{obj}1}$ and $n_{\mathrm{obj}2}$ represent the sizes; $l_{\mathrm{obj}1}$ and $l_{\mathrm{obj}2}$ denote the perimeters; $b_{\mathrm{obj}1}$ and $b_{\mathrm{obj}2}$ are perimeters of the smallest external rectangles; and $n_{\mathrm{Merge}}$, $l_{\mathrm{Merge}}$, and $b_{\mathrm{Merge}}$ are the size, perimeter, and perimeter of the smallest external rectangle of the merged area, respectively.

#### 2.3.2. Reconstitution of Multi-Spectral Imagery

Images with higher spatial and spectral resolution contain richer spatial information. The pixel-based methods are guaranteed to bring more “salt and pepper” noise and may mistake high reflectance land covers in shaded areas as non-shaded. In other words, the information loss of a remote sensing image is mainly caused by the shadows of buildings and viaducts, and the shadows caused by pedestrians, vehicles, and vegetation can be ignored. To apply multi-scale segmentation and using the segmented objects as the minimum processing unit is to ignore the effects of small noise and reduce the interference of high reflectance land covers with small areas while detecting shadow. Rather than processing the segmentation on the RGB model image, all multi-spectral imagery is executed including every band. However, the image after applying multi-scale segmentation is a segmentation map with one band. Therefore, we need to reconstitute the one band image into a multi-spectral image. After the segmentation, we can acquire a layer with similar pixels being merged. Considering a region block in the segmented image as the smallest unit, the value of the pixels in each region were replaced with the mean value of the corresponding pixels in the original image and the replacement was performed in all the bands. Assuming that the total number of segmented regions is n, and each corresponding region contains $k_{i}$ pixels, 1≤i≤n, then the reconstituted image $I_{\mathrm{mr}}$ is:(9)$$I_{\mathrm{mr}}=\overset{}{\underset{b}{\cup }}\overset{n}{\underset{i=1}{\cup }}((\sum_{j=1}^{k_{i}}V_{j})/k_{i}),$$
where *b* is the number of bands and *V* is the value of pixel. The schematic diagram of the mean region is shown in Figure 2.

### 2.4. Shadow Indexes Based on Different Shadow Intensities

#### 2.4.1. Normalized Difference Water Index

In the research of Sun [5], it was verified that the value of the pixels located in a water body were higher than these in shaded regions in the normalized difference water index (NDWI) designed according to the difference of the spectral response between green and NIR channels:(10)$$\mathrm{NDWI}=\frac{\mathrm{green}-\mathrm{NIR}}{\mathrm{green}+\mathrm{NIR}}.$$

In this study, with testing NDWI in both the strong and weak shadow datasets, the same phenomenon was confirmed where the values of water body were higher than those of shadow. The function of NDWI in the four datasets is shown in Figure 3, where water bodies are marked by red rectangles and shadows are marked with green. From the figure, it can be seen that the water bodies were all highlighted with higher values while the shadows had lower values. Unlike the other three datasets, the NDWI values for water bodies and shadows in the Changchun Nov. dataset were closer, which was more likely to cause confusion between water and shadows.

#### 2.4.2. Object-Based Shadow Indexes (OSI)

Because the spectrum of shadows in different images is disparate, the same shadow extraction method cannot be applied to shadows under different illumination intensities. On account of the above situation, the shadow index proposed in this paper consists of three parts: ${\mathrm{OSI}}_{s}$ for strong shadows, ${\mathrm{OSI}}_{w\_WV}$ for weak shadows in the WV-2 and WV-3 data, and ${\mathrm{OSI}}_{w\_GF}$ for weak shadows in the GF-2 data.

In strong shadows, the difference of reflectance between the shaded regions and unshaded regions (except water) is larger than that of weak shadows. Through the above analysis, this study distinguished the low and high reflectance land covers with a darkness index DI, $\mathrm{DI}=1-\mathrm{mean}(\sum_{i=1}^{8}{\mathrm{band}}_{i})$, which takes advantage of the lower mean values of the shadows in all bands. According to spectral statistics, the water bodies and the shadows have smaller values than other land covers in the NIR band; therefore, this study further weakens the effects of other high-reflectivity sites by subtracting NIR from DI. Similarly, because water bodies in NDWI have larger values than shadows, the influence of a water body can be dropped off by using DI minus NDWI. Based on this analysis, the object-based shadow index based on strong shadows is defined as:(11)$${\mathrm{OSI}}_{s}=\left\{ \begin{array}{l} \left(1-\mathrm{mean}\sum_{i=1}^{n}{\mathrm{band}}_{i})-\mathrm{NIR},\;\mathrm{if}\;\mathrm{NIR}\geq (r\times \mathrm{NDWI}\right)\\ (1-\mathrm{mean}\sum_{i=1}^{n}{\mathrm{band}}_{i})-{\mathrm{NDWI}}^{\frac{1}{3}},\;\mathrm{else} \end{array}\right.,$$
where n represents the number of bands and r is the ratio between ambient light intensity and the direct light intensity defined in Section 2.2. Since the value of NDWI is in the range of (−1, 1), ${\mathrm{NDWI}}^{1/3}$ can increase the modulus value of NDWI without changing the positive or negative direction. The function of DI and the impact of ${\mathrm{OSI}}_{s}$ are embodied in Figure 4. By comparing and analyzing what is in the red rectangles, the influence of the water body was reduced significantly. In this case, the same treatment was applied in the weak shadow index to mitigate the effects of the water body, as well as the land covers with high reflectance in NIR.

As for weak shadows, this study improved Sun’s [5] algorithm, which was proved to perform well at shadow extraction for Sentinel-2A MSI imagery. While considering WV-2 and WV-3 data, we used CoastalBule (CB) at 425 nm, NIR2 at 950 nm, green at 545 nm, and NIR1 at 832 nm to substitute for b1 (at 443 nm), b9 (at 945 nm), b3 (at 560 nm), and b8 (at 842 nm) in the shadow enhancement index (SEI), respectively. For the problem of distinguishing shadow from water, we adopted a strategy similar to that of a strong shadow. For the WV-2 and WV-3 imagery, OSI was defined as:(12)$${\mathrm{OSI}}_{w\_\mathrm{WV}}=\left\{ \begin{array}{l} \frac{\left(\mathrm{CB}+\mathrm{NIR}2)-(\mathrm{green}+\mathrm{NIR}1\right)}{\left(\mathrm{CB}+\mathrm{NIR}2)+(\mathrm{green}+\mathrm{NIR}1\right)}-\mathrm{NIR}1,\;\mathrm{if}\;\mathrm{NIR}1\geq (r\times \mathrm{NWIR})\\ \frac{\left(\mathrm{CB}+\mathrm{NIR}2)-(\mathrm{green}+\mathrm{NIR}1\right)}{\left(\mathrm{CB}+\mathrm{NIR}2)+(\mathrm{green}+\mathrm{NIR}1\right)}-\mathrm{NDWI},\;\mathrm{else} \end{array}\right..$$

Unfortunately, the GF-2 data only has the central wavelengths of 485 nm, 557.5 nm, 660 nm, and 830 nm, which cannot be implemented in ${\mathrm{OSI}}_{w\_\mathrm{WV}}$. Therefore, we analyzed the GF-2 data separately for the case of weak shadow and present the shadow index applicable to GF-2 imagery:(13)$${\mathrm{OSI}}_{w\_\mathrm{GF}}=\left\{ \begin{array}{l} \frac{\mathrm{blue}-\mathrm{NIR}}{\mathrm{blue}+\mathrm{NIR}}-\mathrm{NIR},\;\mathrm{if}\;\mathrm{NIR}\geq (r\times NWIR)\\ \frac{\mathrm{blue}-\mathrm{NIR}}{\mathrm{blue}+\mathrm{NIR}}-\mathrm{NDWR},\;\mathrm{else} \end{array}\right..$$

Based on the above analysis, any data containing the green, blue, and NIR bands can enhance the characteristics of shadow by using ${\mathrm{OSI}}_{s}$ in the case of strong shadow. For the weak shadows from the WV-2 and WV-3 data, ${\mathrm{OSI}}_{w\_\mathrm{WV}}$ can highlight and separate shadow from a water body well. Unfortunately, in weak shadows, the spectral information of water and shadow in GF-2 are too similar to be separated; thus, ${\mathrm{OSI}}_{w\_\mathrm{GF}}$ was designed. The flow chart of the proposed method is shown in Figure 5 and the application of the proposed shadow index can be divided into the following steps:
Preprocessing, including radiation calibration, FLAASH atmospheric correction, and image fusion.Calculation of the shadow intensity *r* in the RGB model. If *r* ≥ 4, the image will be processed as strong shadow, and if *r* < 4, the image will be treated as weak shadow in the following steps.Applying multi-scale segmentation and reconstitution of multi-spectral imagery, which will be used as the input image of the following steps.To acquire the NDWI map in which the water bodies have higher values than shadow regions.Applying OSI: the strong shadow image will be executed with ${\mathrm{OSI}}_{s}$, the weak shadow image from WV-2 and WV-3 satellites will be executed with ${\mathrm{OSI}}_{w\_\mathrm{WV}}$, and the weak shadow image from GF-2 satellite will be executed with ${\mathrm{OSI}}_{w\_\mathrm{GF}}$.

## 3. Results

### 3.1. Feasibility Analysis of the Indexes and Cross-Comparison Analysis

From the definition of the shadow intensity in Section 2.2, an image with r≥4 was considered a strong shadow, while an image with r<4 was considered a weak shadow. Therefore, the Tripoli dataset with r=4.95 and the Changchun, Sep. dataset with r=5.5 were processed as having strong shadows for the shadow extraction. Meanwhile, the Hangzhou dataset (r=1.32) and Changchun, Nov. dataset (r=2.1) were regarded as having weak shadows. To better verify the robustness of OSI at different shadow intensities, we counted the mean value of six typical objects (i.e., water body, shadow, vegetation, bare soil, high reflectance, and low reflectance) in OSI to present the separability among shadow and other land covers. The mean values of land covers in strong shadow are provided in Figure 6a, in which the shadow’s mean values were higher than those of other land covers, but the range of the mean values in the Tripoli dataset was wider than that in the Hangzhou, Sep. dataset. Compared with the Changchun, Sep. dataset, the Tripoli dataset was selected from WV-3 data and had more spectral information from 400 nm to 1040 nm. As a result, shadow in the Tripoli dataset was easier to acquire than in the Changchun, Sep. dataset.

Unlike strong shadow, land covers obscured by weak shadow still exhibit its spectral properties, which makes shadow and other land covers, especially water bodies, more difficult to be distinguished. Figure 6b,c reveals that the shadow had its maximum mean value both in the Hangzhou and Changchun Nov. datasets, but the value of the water body and shadow in the Hangzhou dataset were highly similar, which interfered with the shadow extraction. Certainly, shadow could be distinguished well from other regions except the water body, both in the Hangzhou dataset and Changchun Nov dataset.

In addition, this paper also applied ${\mathrm{OSI}}_{s}$ in weak shadow, as well as utilizing ${\mathrm{OSI}}_{w\_\mathrm{WV}}$ and ${\mathrm{OSI}}_{w\_\mathrm{GF}}$ in strong shadow data for cross-comparison analysis, as shown in Figure 7. Figure 7a,c illustrates that ${\mathrm{OSI}}_{s}$ could not enhance the shadow features in weak shadow, indicating that ${\mathrm{OSI}}_{s}$ was no longer applicable to shadow extraction with r≤4. ${\mathrm{OSI}}_{w\_\mathrm{WV}}$ and ${\mathrm{OSI}}_{w\_\mathrm{GF}}$ were generally performed in highlighting strong shadow, as plotted in Figure 7b,d. According to the above discussion, as the shadow spectrum changed with the illumination intensity change, the same shadow extraction algorithm could not be applied to shadows at different shadow intensities.

### 3.2. Performance of the OSI for Strong Shadow

In order to verify the universality and effectiveness of the proposed algorithm, OSI was tested on ten samples from the above four datasets. All the samples contain typical land covers, such as water bodies, roads, vegetation, shadows, and buildings of different scales. Among them, three samples were selected from the Tripoli dataset, two from the Changchun Sep. dataset, three from the Hangzhou dataset, and two from the Changchun Nov. dataset. For the effect of ${\mathrm{OSI}}_{s}$ applied in strong shadow, the images from Figure 8, Figure 9 and Figure 10 were from the process of executing ${\mathrm{OSI}}_{s}$, the shadow extraction results from two Tripoli datasets, and the shadow extraction results from the two Changchun Sep. datasets.

As shown in Figure 8d and Figure 9b,e, ${\mathrm{OSI}}_{s}$ could highlight shadow features and distinguish shadow from water bodies well, and ${\mathrm{OSI}}_{s}$ retained the details of the original image well. Moreover, the shadow extraction results were obtained by setting the threshold (${\mathrm{OSI}}_{s}$ > 0.90), while the shadow areas (Figure 9c,f) were extracted completely, including the small shadows. When using the samples from the Changchun Sep. dataset to verify the OSI, the integrity of the shadow extraction results was still at a high level, indicating that ${\mathrm{OSI}}_{s}$ had a good performance when strengthening the shadow of the remote sensing image.

### 3.3. Performance of the OSI on Weak Shadow

For weak shadow, the land covers in shaded areas can still reflect their spectral features, which makes it more difficult to extract shadow information precisely in weak shadow compared with strong shadow. Therefore, an appropriate strategy for weak shadow extraction is proposed in this study. Instead of using a single index to extract shadow from the WV-3 and GF-2 data in strong shadow, this paper designed different shadow extraction methods for weak shadow from the WV-3 and GF-2 data because there was not as much spectral information in the GF-2 data compared with the WV-3 data. The process of ${\mathrm{OSI}}_{w\_\mathrm{WV}}$ is shown in Figure 11, while the application result and the shadow extraction effects are illustrated in Figure 12c,f. The results manifest the feasibility to judge weak shadow by applying ${\mathrm{OSI}}_{w\_\mathrm{WV}}$ in the WV-2 data.

However, compared with the data from WV-2, the data in GF-2 was short of the spectral information from the CoastalBlue band (central wavelength 425 nm) and NIR2 band (central wavelength 950 nm); therefore, ${\mathrm{OSI}}_{w\_\mathrm{WV}}$ was not applicable. Although ${\mathrm{OSI}}_{w\_\mathrm{GF}}$ could separate shadow from other land covers well (see Figure 13b,e) and get good shadow extraction results (see Figure 13c,f), this did not eliminate interference from the water body, as shown in the rectangle (see Figure 13f).

## 4. Discussion

### 4.1. Comparison between the OSI and Existing Shadow Indexes

#### 4.1.1. The Existing Shadow Indexes Used for Comparison

Scholars have proposed different shadow indexes for multi-spectral remote sensing images at the large scale. This paper compared the latest shadow indexes in recent years with the proposed algorithm. The indexes subject to the comparative analysis were the morphological shadow index (MSI) [49], the combinational shadow index (CSI) [42] for Sentinel-2A MSI imagery, the shadow detection index (SDI) [43], and the normalized saturation-value difference index (NSVDI) [50]. Since the applicable data for these indexes are different, their specific descriptions are shown in Table 3.

#### 4.1.2. Comparison of the Five Indexes

Through the research of this paper, we know that the characteristics and spectral reflectance of shadow are disparate at different illumination intensities. Different shadows cannot be well analyzed under the same index. For the sake of verifying the validity of OSI proposed at different illumination intensities, this paper selected strong and weak shadow data for analysis, and the result comparison of the five shadow indexes are shown in Figure 14, Figure 15 and Figure 16.

In the case of strong shadow, the experimental results from the indexes are depicted in Figure 14 and Figure 16a with the water body as a main disturbance of the shadow extraction. Conclusions for strong shadow can be drawn as follows: (1) according to the red rectangle in Figure 14, OSI and SDI could distinguish a water body from shadow, while MSI, CSI, and NSVDI cannot; (2) the other four indexes, except MSI, performed well regardless of water body, but the details in Figure 14b show that the shadow results of SDI and NSVDI were interfered with more; and (3) the interference of land covers under the shadow in OSI (see Figure 14b) was excluded well due to the multi-scale segmentation processing, and OSI performed best among the five indexes.

When considering weak illumination, the spectral features of the land covers in the shaded regions were more obvious than those in strong shadow such that distinguishing shaded regions and unshaded regions in weak shadow was more difficult than in strong shadow. As shown in Figure 15 and Figure 16b, for weak shadow: (1) the MSI found it difficult to distinguish between low reflectance land covers and shadows, and SDI could not embody the characteristics of shadow in weak shadow and performed the worst among the five indexes; (2) Figure 16 illustrates that the OSI, CSI, and NSVDI could better extract the shadow, while the CSI could not distinguish the water body from shadow; (3) the green boxes in Figure 17b,f show that NSVDI could not distinguish green vegetation from shadow well, and Figure 17b,f reveal that NSVDI was more susceptible to the land covers under shadow than OSI; and (4) OSI performed well at embodying the shadowed characteristics in weak shadow and separated the water body from shadow well.

Since the intensity of shadow in the same location varies greatly for different times, we analyzed the shadows at different illumination intensities for the same location by using the above indexes. Figure 16 illustrates the performance of the three indexes at the same location in both strong and weak shadows in the GF-2 imagery, in which CSI and SDI could not be used due to the lack of bands. In the case of strong shadow (see Figure 16a), OSI and NSVDI could enhance the characteristic of the shadow well, but NSVDI could not eliminate the interference of the water body, which indicates that OSI was highly robust, even with GF-2 data not having as much band information as WV data. Although OSI could highlight shadow in weak shadow in GF-2 data well (see Figure 16a), it could not distinguish between shadow and water body because of the resembling spectral characteristics of shadows and water bodies.

For a more intuitive understanding of the role of the five indexes in shadow extraction, we divided their performance into five levels, as shown in Table 4, including excellent (guaranteed shadow integrity, eliminated salt and pepper noise, and distinguished between shadow and water body), good (enhanced shadow well, guaranteed shadow integrity, eliminated salt and pepper noise, but could not solve water body interference), fair (enhanced shadow characteristic to a certain extent, but could not treat salt and pepper noise nor eliminate water body interference), poor (only distinguished high-reflective land covers from low-reflective landcovers), and invalid (an invalid index cannot be used).

### 4.2. Evaluation of the Shadow Extraction Accuracy

To verify the shadow extraction accuracy of the five indexes, the threshold segmentation was performed on the result of each index by judging the value of the pixels larger than the threshold as shadow. While using the Tripoli dataset and the Hangzhou dataset to calculate the five shadow indexes, we adjusted the threshold to obtain the best shadow segmentation result of each index. The shadow segmentation threshold for each index is listed in Table 5. The pixels with the values larger than the threshold were judged as a shaded region, whereas the pixels with the values smaller than the threshold were judged as an unshaded region. The shadow extraction results of the five indexes at different shadow intensities are plotted in Figure 17, Figure 18 and Figure 19.

After extracting the shadow with each shadow index, the quantitative analysis of the shadow extraction results in Figure 17, Figure 18 and Figure 19 were applied. The producer’s accuracy (PA), user’s accuracy (UA), overall accuracy (OA), and kappa coefficient were calculated using a confusion matrix were used to evaluate the accuracy of the shadow extraction, where OA and PA were used to evaluate the accuracy of the shadow detection, and OA and the kappa coefficient were used to assess the accuracy of discrimination between the shadow and the background [6]. Since the experimental data from the Tripoli dataset contained a large area of water, if the accuracy of water extraction was added to the shadow accuracy evaluation, the accuracy of MSI and NSVDI (which could not distinguish shadow from water) would be greatly reduced. Therefore, the water in the Tripoli dataset was ignored when evaluating the shadow extraction accuracy. The accuracies of the shadow extraction results of the shadow indexes in strong shadow are given in Table 6. When evaluating the accuracy of the shadow extraction using the Hangzhou dataset, this paper took water into account by considering the water body as a non-shadow area. The accuracy of the shadow extraction results of the weak shadow in the WV data are shown in Table 7. Since SDI and CSI could not be used for GF-2 data, this paper only evaluated the shadow extraction accuracies of OSI, MSI, and NSVDI. In addition, since all shadow extraction methods could not separate shadow from the water body in the GF-2 weak shadow data, the water body was neglected from the calculation of the shadow extraction accuracy, and the accuracy of the shadow extraction results are listed in Table 8.

Regardless of the influence of the water body, it can be seen from Figure 17 and Table 6 that MSI could not distinguish shadow from land covers with low spectral reflectance (dark roofs, asphalt roads, etc.) and misinterpreted many unshaded regions as shadow, leading to the kappa coefficient of 0.1506 and the OA of 53.47%. The performance of CSI was fair for shadow extraction (PA: 35.78% and UA: 51.88%), and discrimination (OA: 74.17% and kappa: 0.2643) between shadow and background in strong shadow. The shadow extraction results of SDI were similar to those of CSI, where both of which could distinguish between shadow and the water body (see Figure 17d,e). However, SDI (kappa: 0.5050) performed better than CSI (kappa: 0.2643) on partition shadows and backgrounds. Separating shadows from backgrounds (OA: 89.47% and kappa: 0.7444), NSVDI could not distinguish the water body from shadow (see Figure 17f). The PA (97.00%) and UA (96.62%) of OSI were both above 95%, indicating that OSI could extract shadow well (see Figure 17b) from the Tripoli dataset, which was considered to be strong shadow. Furthermore, OSI performed almost perfectly regarding discrimination between the shadows and backgrounds (OA: 98.30% and kappa: 0.9565). In summary, when the image was in strong shadow, the performance of the five indexes with water body ignored was ordered as follows: OSI > NSVDI > SDI > MSI > CSI.

Since the Hangzhou dataset contained a large number of land covers with lower spectral reflectance, MSI (see Figure 18c) misjudged a large number of asphalt roads and water bodies as shadow areas, though it was moderately able to extract shadows (PA: 73.99% and UA: 58.25%), which means that MSI had the slight ability in discriminating between shadow and background (OA: 55.95% and kappa: 0.0749). The CSI could substantially detect shadow (PA: 75.67% and UA: 89.37%), as well as distinguishing between water body and shadow (OA: 95.67 and kappa: 0.7952), but CSI (see Figure 18d) found it easier to produce small noises than the OSI (see Figure 18b). In the case of weak shadow, the SDI could not reflect the shadow characteristics at all, so we treated it as invalid. The results of NSVDI were similar to, but weaker than, those of CSI (see Figure 18d,e). OSI also had a perfect performance in discrimination between shadow and background (OA: 98.71% and kappa: 0.9412) and the PA and OA of OSI were both above 90%. According to the performance, the five indexes were ordered as follows: OSI > CSI > NSVDI > MSI > SDI.

As for the weak shadow in the Changchun, Nov dataset, CSI and SDI could not be used anymore. From Figure 19c and Table 8, it was shown that MSI had a moderate effect regarding shadow extraction (OA: 73.99% and PA: 85.28%), as well as a slight effect of discrimination between shadow and background (OA: 55.95% and kappa: 0.0749). As shown in Figure 19d, NSVDI manifested a moderate performance regarding both shadow detection and distinguishing between shadow and background. Similar to the results and accuracies in strong and weak shadows in the WV data, OSI still performed well on the GF-2 weak shadow data, including extracting shadows (PA: 95.51% and UA: 98.34%), and in distinguishing between shadow and background, with an OA of 95.78% and kappa of 0.9148. In conclusion, the performance of the indexes in weak shadow on the GF-2 data was ordered as follows: OSI > NSVDI > MSI.

According to the quantitative evaluation of the shadow extraction results of the five indexes at different shadow intensities, we know that the CSI had a huge difference in shadow detection at different shadow intensities and was not available in the GF-2 data in weak shadow. NSVDI could extract shadow regions better but could not distinguish between the shadow and water body. SDI provided the opposite results in shadow with different illumination intensities and was completely invalid in the case of weak shadow. For both strong and weak shadows, OSI could distinguish shadow from non-shadow, and exclude the water body interference in shadow extraction well, except for the water in the GF-2 data in weak shadow. In order to see the performance of each index more intuitively, we summarize the details for each index in Table 9.

## 5. Conclusions

The purpose of this research was to propose a shadow extraction method that is suitable for high-resolution multispectral remote sensing images at a large scale. According to the analysis of the existing shadow extraction methods in the Introduction (Section 1), it is known that the index method is the most effective method for city-scale shadow extraction. Other methods that also extract shadow information well can only be applied in small-scale images due to their complexity. Although many shadow indexes have been proposed at present, we find that the spectral variation of shadows under different illimitation conditions is so large that the same index cannot be applied to different shadow conditions. Therefore, this study proposed a shadow index OSI that is suitable for different illumination intensities. According to the illumination intensity *r*, this study treated images with r≥4 as being strong shadow, and images with r<4 as being weak shadow. In order to improve the integrity of the shadow extraction results and eliminate small noises, this study reconstructed the image after multi-scale segmentation. From the shadow extraction results in Figure 17, Figure 18 and Figure 19, image reconstruction strategy reduced the noises well and improved the accuracy of the shadow extraction. Based on different data sources and shadow intensities, OSI was designed in three parts: ${\mathrm{OSI}}_{s}$ for strong shadows, ${\mathrm{OSI}}_{w\_\mathrm{WV}}$ for weak shadows from the WV data, and ${\mathrm{OSI}}_{w\_\mathrm{GF}}$ for weak shadows from the GF-2 data. With the purpose to eliminate the influence of the water body on shadow detection, NDWI and NIR were used to separate shadow from water body and low reflectance land covers.

To prove the effectiveness and robustness of OSI, this paper conducted quantitative and qualitative analysis of the performance of OSI and compared it with four recently proposed shadow indexes. The performances of OSI in strong and weak shadows were analyzed qualitatively in Section 3.2 and Section 3.3, and the results show that OSI could enhance shadow and separate shadow from water body effectively (except water body in weak shadow from the GF-2 data). In Section 4.2, this paper posts quantitative analysis of OSI and the other four shadow indexes, and the experimental results demonstrated that OSI provided the highest shadow extraction accuracy in both strong and weak shadows.

The OSI was mainly restricted by its inability to separate water body and shadow in weak shadow from the GF-2 data, which was attributed to its lack of rich spectral information. The spectral analysis revealed that the spectral curves of water body and shadow were almost the same in the weak shadows from the GF-2 data. Consequently, it was almost impossible to use the proposed index method to separate water body and shadow in that case, and we are trying to find a new method to overcome such limitation.

The proposed shadow index has shown its potential for shadow detection in different illumination intensities at a larger scale. For the future research directions, we will focus on extending the application scope of the index, as well as eliminating the influence of water bodies in GF-2 images with weak shadow. Furthermore, we are trying to use the shadows extracted by the index to recover the information of shaded areas on a large scale.

## Figures and Tables

**Figure 1 sensors-20-01077-f001:**
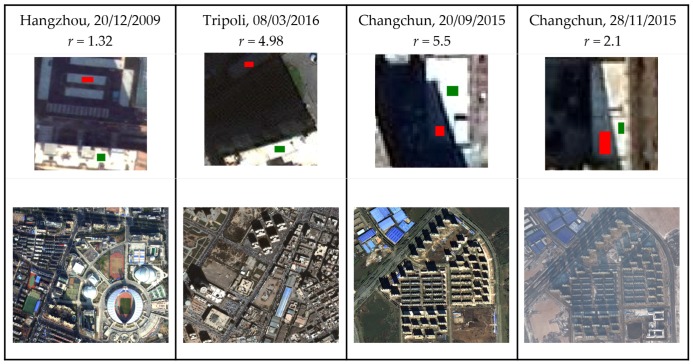
The shadow intensity calculation of the four datasets, where the images in the first row selected for the *r* calculations are the details of the four datasets and the images in the second row are the shadow situations of the datasets.

**Figure 2 sensors-20-01077-f002:**
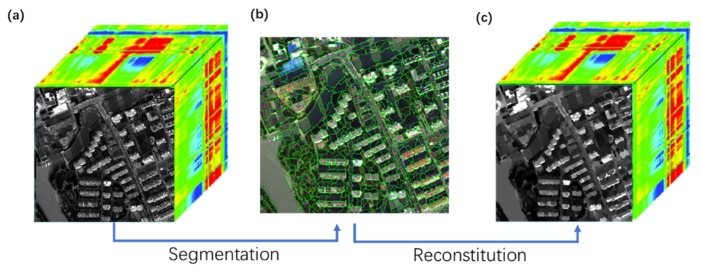
The method of reconstitution of multi-spectral imagery: (**a**) original image with multi-bands, (**b**) the image after being segmented (the green lines are the boundaries of each region), and (**c**) the reconstituted image.

**Figure 3 sensors-20-01077-f003:**
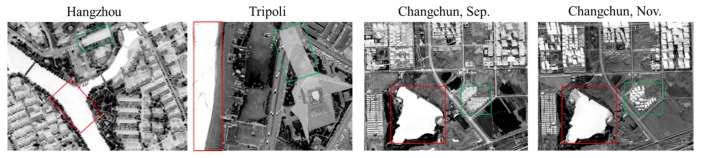
NDWI of the four datasets: the water bodies (in red rectangles) were more prominent than the shadows (in green rectangles).

**Figure 4 sensors-20-01077-f004:**
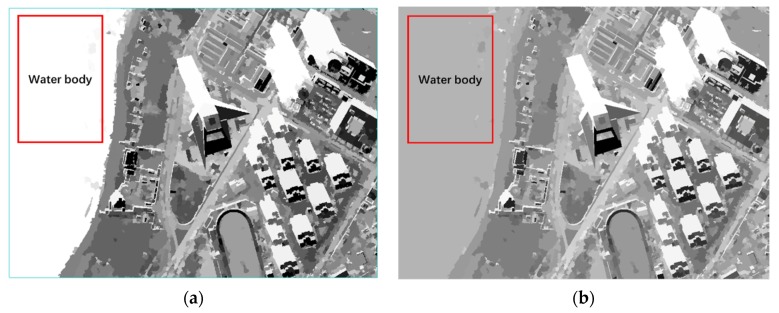
The abridged general view showing the step of mitigating the effect of the water body in the Tripoli dataset: (**a**) the result using the darkness index (DI) with the water body and shadow both highlighted, and (**b**) the result using ${\mathrm{OSI}}_{s}$ with the effect of the water body mitigated.

**Figure 5 sensors-20-01077-f005:**
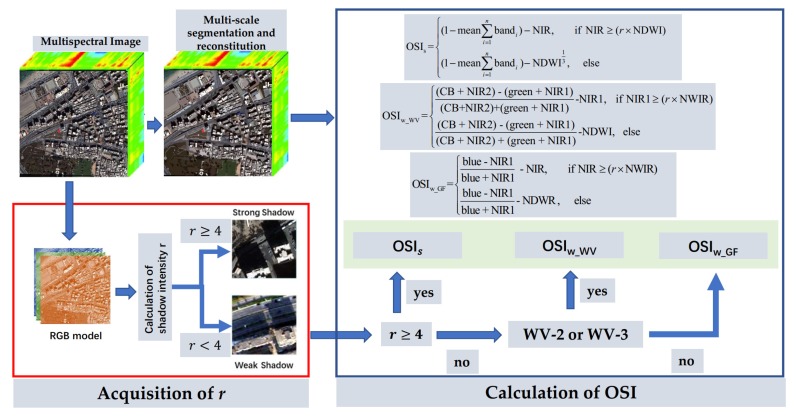
Flowchart of the proposed object-based shadow index (OSI).

**Figure 6 sensors-20-01077-f006:**
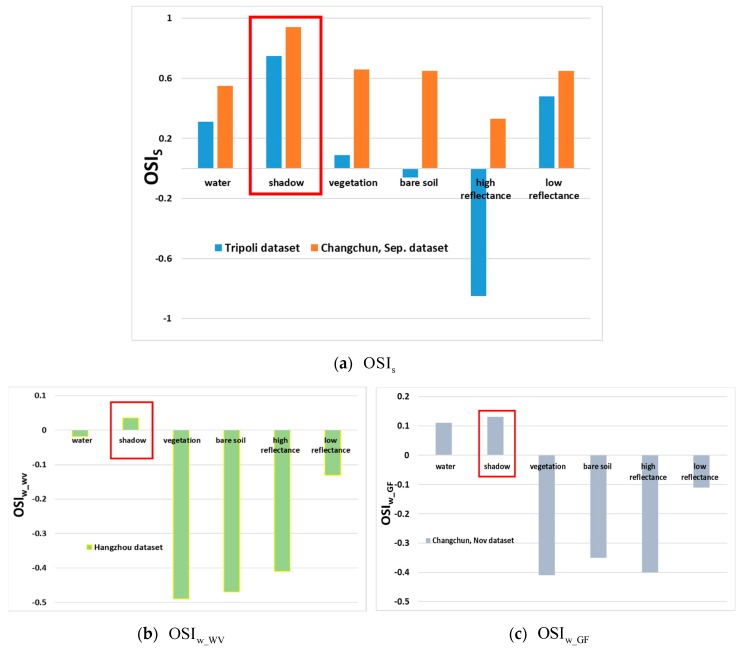
Mean values of the six typical land covers in OSI with shadows marked by red rectangles; the shadow has its maximum mean value in the six land covers of the four datasets. (**a**) Statistics of strong shadow in the Tripoli and Changchun, Sep. datasets. (**b**) Statistics of weak shadow in the Hangzhou dataset. (**c**) Statistics of weak shadow in the Changchun Nov. dataset.

**Figure 7 sensors-20-01077-f007:**
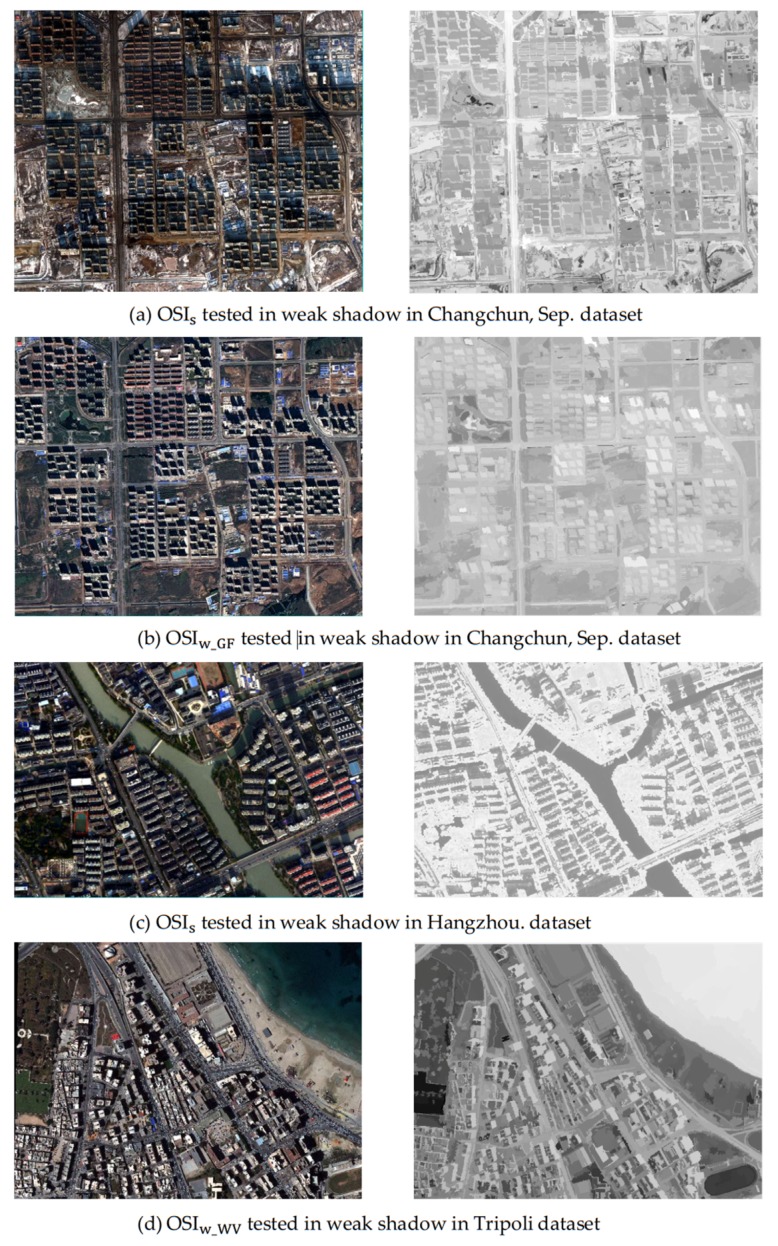
The cross-comparison analysis: ${\mathrm{OSI}}_{s}$ designed for strong shadow could not highlight shadow in the weak shadow datasets; ${\mathrm{OSI}}_{w\_\mathrm{WV}}$ could not distinguish shadow from water body in strong shadow; and ${\mathrm{OSI}}_{w\_\mathrm{GF}}$ was unable to highlight shadow in strong shadow.

**Figure 8 sensors-20-01077-f008:**
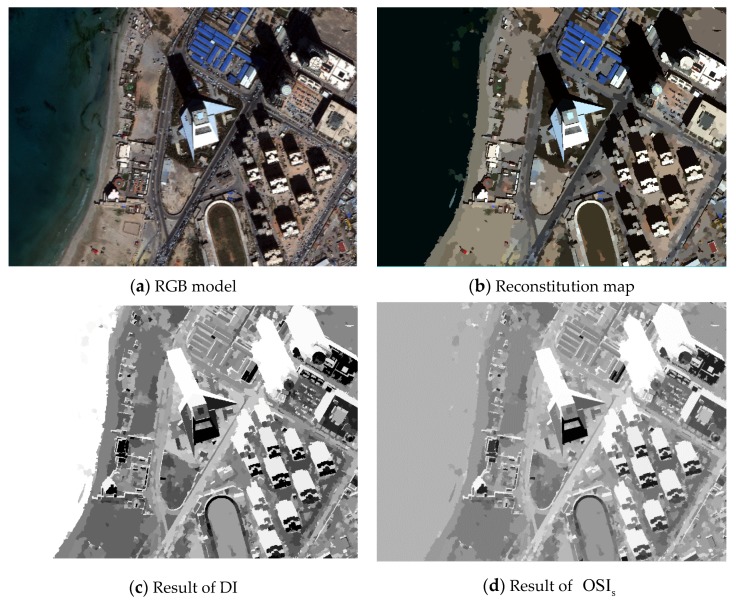
The process of applying ${\mathrm{OSI}}_{s}$ in strong shadow for the Tripoli dataset.

**Figure 9 sensors-20-01077-f009:**
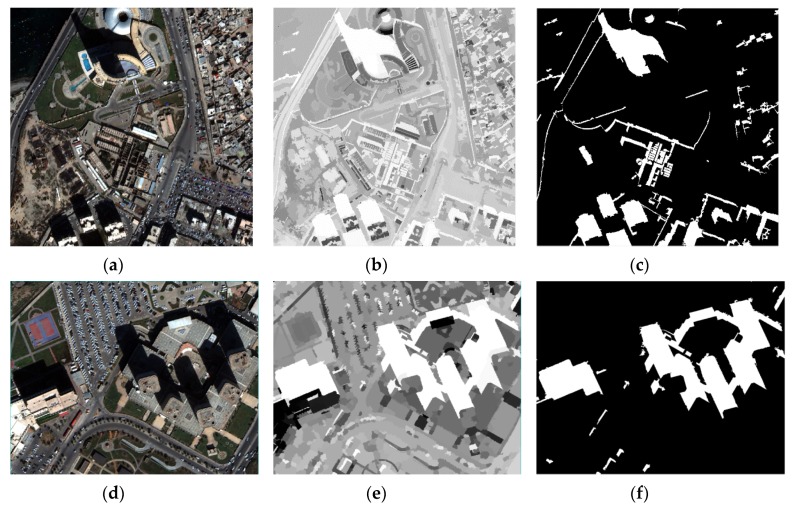
Tested using the Tripoli dataset (strong shadow) for shadow extraction: (**a**,**d**) original images, (**b**,**e**) the results of applying ${\mathrm{OSI}}_{s}$, and (**c**,**f**) the results of shadow extraction when setting ${\mathrm{OSI}}_{s}\geq 0.90$ (satisfactory shadow extraction results and elimination of the water body influence).

**Figure 10 sensors-20-01077-f010:**
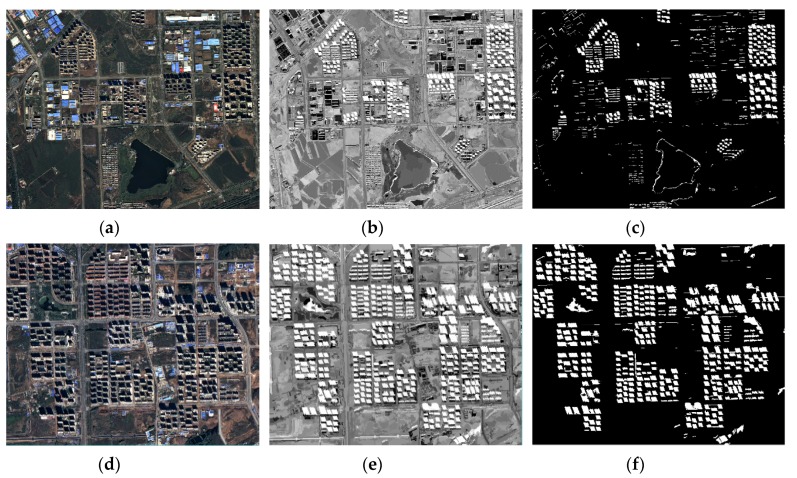
${\mathrm{OSI}}_{s}$ tested on the Changchun Sep. dataset (strong shadow) for shadow extraction: (**a**,**d**) original images, (**b**,**e**) the results of applying ${\mathrm{OSI}}_{s}$, and (**c**,**f**) the results of shadow extraction when setting ${\mathrm{OSI}}_{s}\geq 0.70$ (satisfactory shadow extraction results and elimination of the water body influence).

**Figure 11 sensors-20-01077-f011:**
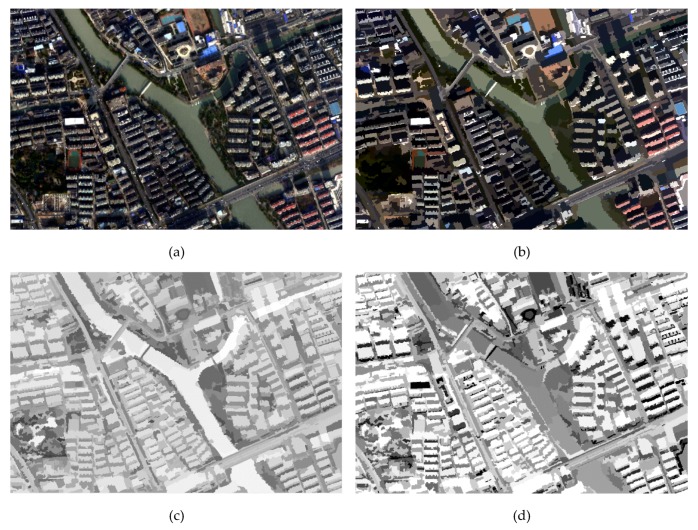
The process of applying ${\mathrm{OSI}}_{w\_\mathrm{WV}}$ in strong shadow for the Hangzhou dataset: (**a**) original image from the Hangzhou dataset displayed in the RGB model, (**b**) the multi-scale segmentation map, (**c**) the result before applying the water body weakened strategy, and (**d**) the result of utilizing ${\mathrm{OSI}}_{w\_\mathrm{WV}}$.

**Figure 12 sensors-20-01077-f012:**
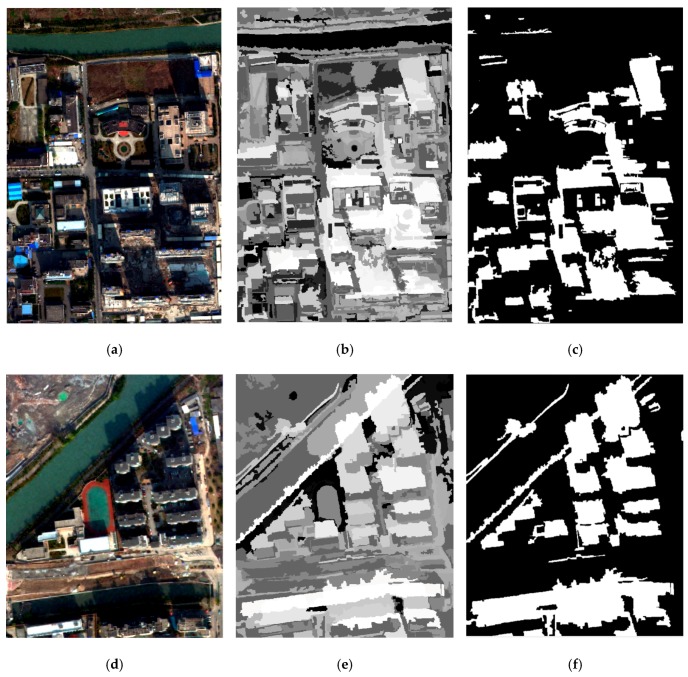
${\mathrm{OSI}}_{w\_\mathrm{WV}}$ tested on the Hangzhou dataset: (**a**,**d**) original image shown in an RGB combination, (**b**,**e**) the results of applying ${\mathrm{OSI}}_{w\_\mathrm{WV}}$, and (**c**,**f**) the results of shadow extraction when setting ${\mathrm{OSI}}_{w\_\mathrm{WV}}\geq 0.13$ (satisfactory shadow extraction results and elimination of the water body influence).

**Figure 13 sensors-20-01077-f013:**
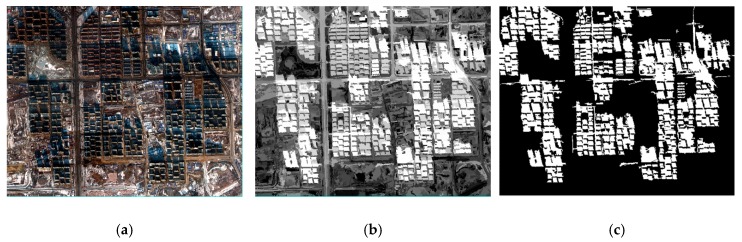
${\mathrm{OSI}}_{w\_\mathrm{GF}}$ tests in weak shadow: (**a**,**d**) the original images, (**b**,**e**) the results of applying ${\mathrm{OSI}}_{w\_\mathrm{GF}}$, and (**c**,**f**) the results of shadow extraction when setting ${\mathrm{OSI}}_{w\_\mathrm{GF}}\geq -0.035$ (satisfactory shadow extraction results but one water body was mistaken for shadow, as shown in the red rectangle).

**Figure 14 sensors-20-01077-f014:**
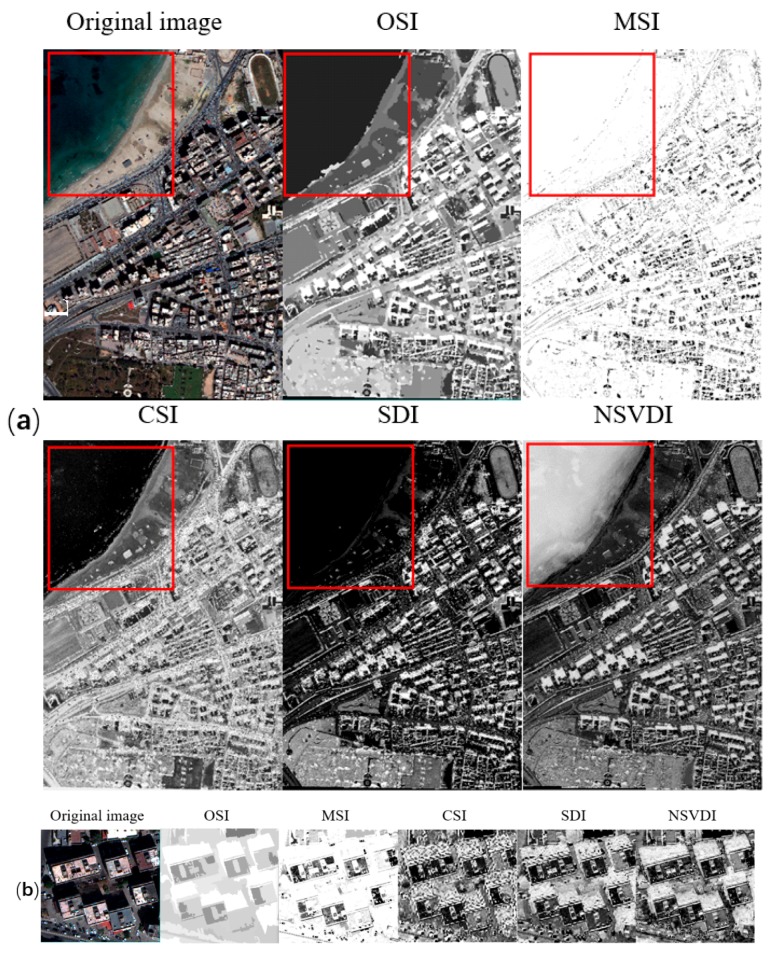
Comparison of the five indexes in strong shadow using the data from the Tripoli dataset: (**a**) the results of the five indexes with the water body marked using a red rectangle, and (**b**) the performance of the five indexes shown in details.

**Figure 15 sensors-20-01077-f015:**
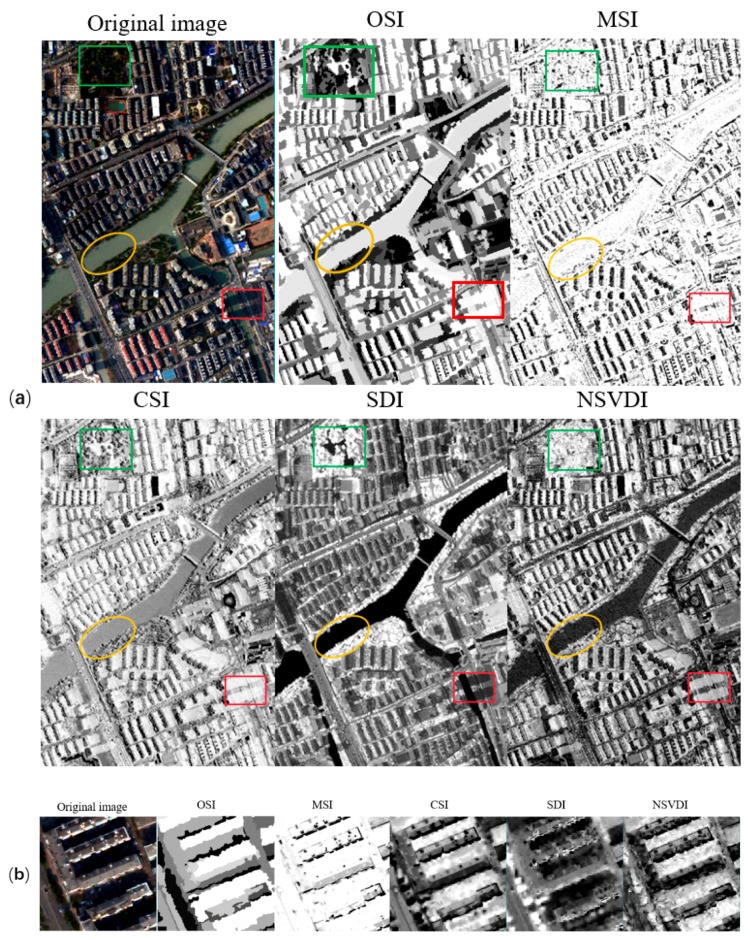
Comparison of the five indexes in strong shadow using the data from the Hangzhou dataset: (**a**) the results of the five indexes with the water body marked using a yellow ellipse, shadow marked using a red rectangle, and vegetation marked using a green rectangle; and (**b**) the performance of the five indexes shown in detail.

**Figure 16 sensors-20-01077-f016:**
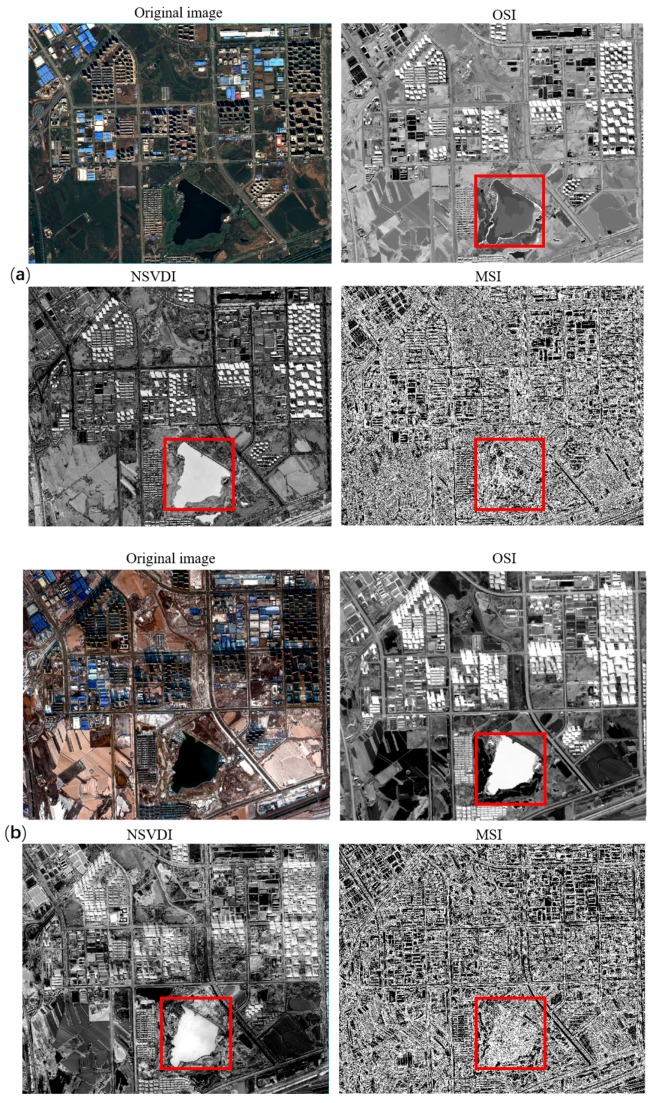
Comparison of three indexes in the same location at different shadow intensities: (**a**) the results of the three indexes in strong shadow with the water body marked using a red rectangle, and (**b**) the result of the three indexes in weak shadow with the water body marked using a red rectangle.

**Figure 17 sensors-20-01077-f017:**
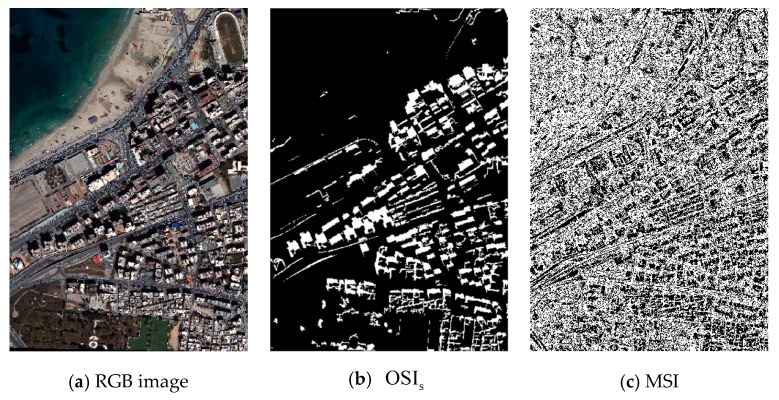
Shadow extraction results of the five indexes using the Tripoli dataset (strong shadow).

**Figure 18 sensors-20-01077-f018:**
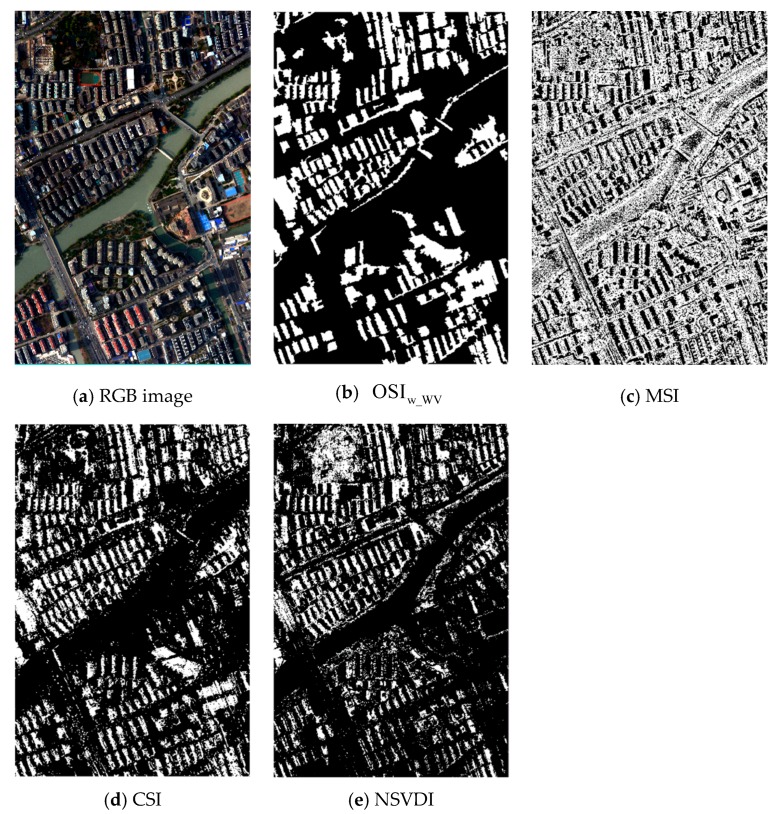
Shadow extraction results of the indexes, except SDI which was invalid, using the Hangzhou dataset (weak shadow).

**Figure 19 sensors-20-01077-f019:**
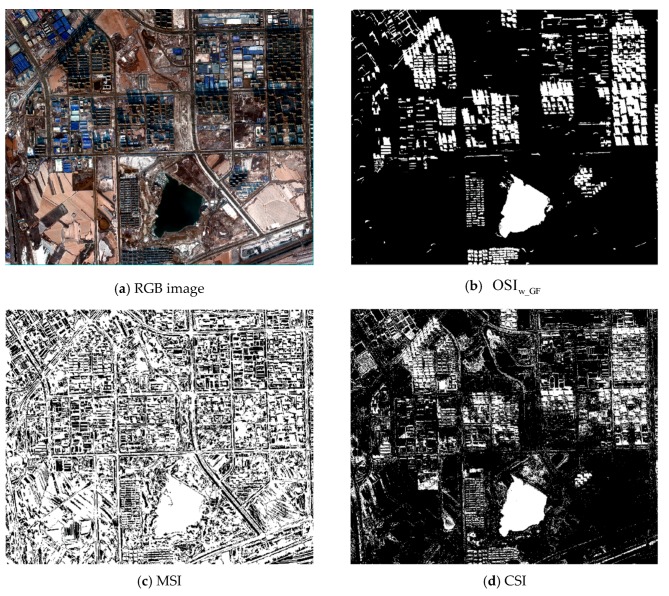
Shadow extraction results of OSI, MSI, and NSVDI using the Changchun, Nov dataset (weak shadow).

**Table 1 sensors-20-01077-t001:** Details of GaoFen-2 (GF-2) imagery.

Bands	Blue	Green	Red	NIR
Central Wavelength (nm)	485	557.5	660	830
Resolution	Multispectral resolution: 3.24 m Panchromatic resolution: 0.81 m			
Location	Changchun, China			
Local time	20/09/2015, 10:42 28/11/2015, 10:44			

**Table 2 sensors-20-01077-t002:** Details of WorldView-2 (WV-2) and WorldView-3 (WV-3) imagery.

Bands		CoastalBlue	Blue	Green	Yellow	Red	RedEdge	NIR1	NIR2
Central Wavelength (nm)		425	480	545	605	660	725	832.5	950
Resolution	WorldView-2	Multispectral resolution: 1.84 m							
		Panchromatic resolution: 0.46 m							
	WorldView-3	Multispectral resolution: 1.24 m							
		Panchromatic resolution: 0.31 m							
		Short-wavelength infrared: 3.7 m							
Local time	Hangzhou	20/12/2009, 10:48							
	Tripoli	08/03/2016, 12:12							

**Table 3 sensors-20-01077-t003:** The definition of the indexes and parameters setting.

Index	Data	Definition	Description
MSI	WorldView-2	$\mathrm{MSI}=\frac{\sum_{s\in S}\sum_{d\in D}{\mathrm{DMP}}_{\mathrm{BTH}(d,s)}}{N_{D}\times N_{S}}$, $BTH=\gamma_{b}^{re}(d,s)-b$ where *b* is the brightness image, obtained using a morphological opening operation on *b*; *s* and *d* are the size and direction of the Black top-hat (BTH) transformation, respectively; and $N_{S}$ and $N_{D}$ are the total number and total direction, respectively.	Parameters set in our data: s∈(2:5:52), $d\in (0^{\circ }:45^{\circ }:180^{\circ })$.
CSI	Sentinel-2A	$\mathrm{SEI}=\frac{{(b}_{443\mathrm{nm}}+b_{945\mathrm{nm}})-(b_{560\mathrm{nm}}+b_{842\mathrm{nm}})}{{(b}_{443\mathrm{nm}}+b_{945\mathrm{nm}})+(b_{560\mathrm{nm}}+b_{842\mathrm{nm}})}$ $\mathrm{CSI}=\left\{ \begin{array}{l} \mathrm{SEI}-\mathrm{NIR},\;\mathrm{if}\;\mathrm{NIR}\geq \mathrm{NDWI}\\ \mathrm{SEI}-\mathrm{NDWI},\;\mathrm{else} \end{array}\right.$	Bands selection in WV-2 and WV-3 data: $\mathrm{SEI}=\frac{\left(\mathrm{CoastalBlue}+\mathrm{NIR}2)-(\mathrm{Green}+\mathrm{NIR}1\right)}{\left(\mathrm{CoastalBlue}+\mathrm{NIR}2)+(\mathrm{Green}+\mathrm{NIR}1\right)}$ $\mathrm{CSI}=\left\{ \begin{array}{l} \mathrm{SEI}-\mathrm{NIR}1,\;\mathrm{if}\;\mathrm{NIR}\geq \mathrm{NDWI}\\ \mathrm{SEI}-\mathrm{NDWI},\;\mathrm{else} \end{array}\right.$ Not applicable for GF-2 dataset.
SDI	WorldView-2	$\mathrm{SDI}=\frac{\mathrm{NIR}2-\mathrm{Blue}}{\mathrm{NIR}2+\mathrm{Blue}}-\mathrm{NIR}1$	Not applicable for GF-2 dataset.
NSVDI	IKONOS	$\mathrm{NSDVI}=\frac{S-V}{S+V}$,	R, G, and B bands were selected for combination. Then, the RGB model image was converted from the RGB model to the hue-saturation-value (HSV) space.

**Table 4 sensors-20-01077-t004:** Performance of the five indexes with qualitative analysis.

Dataset	Shadow Intensity	OSI	MSI	CSI	SDI	NSVDI
Tripoli	Strong	Excellent	Poor	Fair	Good	Good
Hangzhou	Weak	Excellent	Poor	Good	Poor	Fair
Changchun, Sep.	Strong	Excellent	Poor	Invalid	Invalid	Good
Changchun, Nov.	Weak	Good	Poor	Invalid	Invalid	Fair

**Table 5 sensors-20-01077-t005:** Threshold selected for different shadow intensities.

Dataset	OSI	MSI	CSI	SDI	NSVDI
Tripoli	0.92	0	0	1	0.4
Hangzhou	0.13	0	−0.1	Invalid	0.3
Changchun, Nov	−0.03	0	Unavailable	Unavailable	0.3

**Table 6 sensors-20-01077-t006:** Comparison of the shadow extraction accuracy of the five indexes in strong shadow (quantitative analysis of Figure 17).

	Accuracies			
Index	PA (%)	UA (%)	OA (%)	Kappa
MSI	75.74	33.37	53.47	0.1506
CSI	35.78	51.88	74.17	0.2643
SDI	48.01	80.95	83.22	0.5050
NSVDI	89.02	75.60	89.47	0.7444
${\mathrm{OSI}}_{s}$	97.00	96.62	98.30	0.9565

**Table 7 sensors-20-01077-t007:** Comparison of the shadow extraction accuracy of the five indexes in weak shadow (quantitative analysis of Figure 18).

	Accuracies			
Index	PA (%)	UA (%)	OA (%)	Kappa
MSI	76.43	15.69	43.60	0.0572
CSI	75.67	89.37	95.67	0.7952
SDI	Invalid			
NSVDI	49.70	80.73	91.92	0.5730
${\mathrm{OSI}}_{w\_\mathrm{WV}}$	91.61	98.34	98.71	0.9412

**Table 8 sensors-20-01077-t008:** Comparison of the shadow extraction accuracy of the five indexes in weak shadow (quantitative analysis of Figure 19).

	Accuracies			
Index	PA (%)	UA (%)	OA (%)	Kappa
MSI	73.99	58.25	55.95	0.0749
CSI	Unavailable			
SDI	Unavailable			
NSVDI	57.28	86.00	70.99	0.4365
${\mathrm{OSI}}_{w\_\mathrm{WV}}$	95.51	98.34	95.78	0.9148

**Table 9 sensors-20-01077-t009:** Summary of performances of the five indexes.

Index	Advantages	Disadvantages
MSI	Applicable for all datasets.	A slight effect in discriminating between shadow and background; unable to distinguish shadow from water body; the lowest accuracy among the five indexes.
CSI	Able to detect shadow in weak shadow well; able to distinguish shadow from water body.	Not applicable for GF-2 data; not applicable for strong shadow.
SDI	Good performance in strong shadow.	Not applicable for GF-2 data; the worst preference in weak shadow among the indexes.
NSVDI	Able to detect shadow moderately both in strong and weak shadows; applicable for all datasets.	Unable to distinguish shadow from water body in all datasets.
OSI	The best performance among the indexes with the highest accuracy in all conditions; applicable for all multi-spectral data; able to distinguish shadow from the water body in most multi-spectral data.	Unable to distinguish shadow from water body in the GF-2 dataset with weak shadow.

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
