# Peer review of "Object-Based Shadow Index via Illumination Intensity from High Resolution Satellite Images over Urban Areas"

_sensors, 2020, doi:10.3390/s20041077_

Round 1

Reviewer 1 Report

This paper deals with an interesting problem of shadow recognition in high-resolution urban imagery from satellites. Its current quality of presentation, however, deters the readers from appreciating the authors' research findings.

It is strongly advised that authors check extensively on their vocabulary choice, sentence structures, as well as manuscript formats. First, please remove strikethroughs and underlines (in blue and black colors) to leave only the final version of the document. Second, refrain from using subjective or unprofessional modifiers, including 'like(such as),' 'excellent,' 'enormous,' etc. Third, the reference should be reformatted now that the journal names seem to be mingled with the initials of authors' names. Use supporting data instead.

I will not list all typographical errors here, but the following may serve as a starting point for the revision process.

Lines 54-55: are absent -> lack

59-61: wrong sentence

89-92: According to -> In order to

143: Any preprocessing on imagery? (e.g. cloud removal) If so, please describe it.

162: I=Ie instead of I=Id?

297: Fig 5 text is too small.

420: Science -> Since -> Because

Author Response

Dear reviewer:

Firstly, thank you very much for your comments concerning our manuscript. Those comments are all valuable and very helpful for revising and improving our paper, as well as the important guiding significance to our researches. We have modified each suggestion accordingly, and I believe the article has been comprehensively improved. We very much hope that our revision could satisfy you. If there are any other mistakes or improper writing, please inform us and we will modify carefully. Finally, look forward to hearing from you soon.

Point 1: This paper deals with an interesting problem of shadow recognition in high-resolution urban imagery from satellites. Its current quality of presentation, however, deters the readers from appreciating the authors' research findings.

It is strongly advised that authors check extensively on their vocabulary choice, sentence structures, as well as manuscript formats. First, please remove strikethroughs and underlines (in blue and black colors) to leave only the final version of the document. Second, refrain from using subjective or unprofessional modifiers, including 'like (such as),' 'excellent,' 'enormous,' etc. Third, the reference should be reformatted now that the journal names seem to be mingled with the initials of authors' names. Use supporting data instead.

Response 1: We are so sorry for our unprofessional English writing and inadequate result analysis! After carefully check, we found many grammar and sentence errors, and have modified the manuscript accordingly. If there are still any problems, please give us a chance and we will modify timely.

Point 2: I will not list all typographical errors here, but the following may serve as a starting point for the revision process.

Lines 54-55: are absent -> lack 59-61: wrong sentence 89-92: According to -> In order to 162: I=Ie instead of I=Id? 420: Science -> Since -> Because

Response 2: Thanks for the careful attention to the details of my article. We have modified them accordingly.

Point 3: 143: Any preprocessing on imagery? (e.g. cloud removal) If so, please describe it.

Response 3: Thank you for your reminding, the pre-processing is necessary. We have added the description of pro-processing and the data choice at the last paragraph of Section 2.1.

Point 4: Fig 5 text is too small.

Response 4: We have rewritten the text in the Figure 5 and we hope it clearly expresses the research process of this paper.

Reviewer 2 Report

The manuscript contains a correction made by the Authors. It should be stressed that the reviewer has not reviewed the first version (or previous versions) of this article.

Comments formulated during my review are presented below. These are as follows:

1) It should be noted that the optical or infrared instruments traditionally used for remote sensing are unfortunately compromised by the effects of
cloudiness and impacted by the access of sunlight. When using the Synthetic Aperture Radar (SAR) technology, this problem does not occur.

The Synthetic Aperture Radar produces high-resolution images throughout its operation and regardless of weather conditions. Because of these advantages, the SAR technology is widely used in remote sensing applications for Earth observations.

The Authors should also avoke papers dealing with image segmentation problems in remote sensing using SAR technology, namely:

[a] "Superpixel Segmentation of Polarimetric Synthetic Aperture Radar (SAR) Images Based on Generalized Mean Shift". Remote Sensing, 2018,
vol 10(10), 1592.
[b] "River channel segmentation in polarimetric SAR images: watershed transform combined with average contrast maximisation."
Expert Systems with Applications, 2017, vol. 82, 196-215.
[c] "A Median regularized level set for hierarchical segmentation of SAR images". IEEE Geoscience and Remote Sensing Letters, 2017, vol. 14(7), 1171-1175.

[d] "Level Set Segmentation Algorithm for High-Resolution Polarimetric SAR Images Based on a Heterogeneous Clutter Model."IEEE Journal of Selected Topics in Applied Earth Observations and Remote Sensing, 2017, vol. 10(10), 4565-4579.

2) What is the novelty of this work? This should be clearly mentioned and highlighted in the paper.

3) The images in Figure 1 are blurred. Can Authors place better quality images (e.g. higher resolution?)? The authors state that the images come from September and November. What year?

4) In the related work section, a more rigorous investigation on the existing methods, such as comparison of previous approaches in terms of pros
and cons, should be given. A summary table can be used in this regard.

5) The Authors need to present and discuss several solid future research directions.

Author Response

Dear reviewer:

Firstly, thank you very much for your comments concerning our manuscript. Those comments are all valuable and very helpful for revising and improving our paper, as well as the important guiding significance to our researches. We have modified each suggestion accordingly, and I believe the article has been comprehensively improved. We very much hope that our revision could satisfy you. If there are any other mistakes or improper writing, please inform us and we will modify carefully. Finally, look forward to hearing from you soon.

The responses are as follows:

Point 1: It should be noted that the optical or infrared instruments traditionally used for remote sensing are unfortunately compromised by the effects of cloudiness and impacted by the access of sunlight. When using the Synthetic Aperture Radar (SAR) technology, this problem does not occur.

The Synthetic Aperture Radar produces high-resolution images throughout its operation and regardless of weather conditions. Because of these advantages, the SAR technology is widely used in remote sensing applications for Earth observations.

Response 1: Thank you for reminding me of that the SAR technology doesn’t be affected by solar illumination conditions and we’ve added some description in the abstract and introduction.

Point 2: The Authors should also avoke papers dealing with image segmentation problems in remote sensing using SAR technology, namely:

[a] "Superpixel Segmentation of Polarimetric Synthetic Aperture Radar (SAR) Images Based on Generalized Mean Shift". Remote Sensing, 2018,

vol 10(10), 1592.

[b] "River channel segmentation in polarimetric SAR images: watershed transform combined with average contrast maximisation."

Expert Systems with Applications, 2017, vol. 82, 196-215.

[c] "A Median regularized level set for hierarchical segmentation of SAR images". IEEE Geoscience and Remote Sensing Letters, 2017, vol. 14(7), 1171-1175.

[d] "Level Set Segmentation Algorithm for High-Resolution Polarimetric SAR Images Based on a Heterogeneous Clutter Model."IEEE Journal of Selected Topics in Applied Earth Observations and Remote Sensing, 2017, vol. 10(10), 4565-4579.

Response 2: Thank you for your suggestion and we have carefully reading the above references. The above segmentation methods are also suitable for multi-spectral image segmentation and we have referenced them in this article in the introduction.

Point 3: What is the novelty of this work? This should be clearly mentioned and highlighted in the paper.

Response 3: Thanks for pointing out the shortcoming. We have highlighted the novelty at the last paragraph of the introduction.

Point 4: The images in Figure 1 are blurred. Can Authors place better quality images (e.g. higher resolution?)? The authors state that the images come from September and November. What year?

Response 4: We have re-drawn Figure 1 with adding better quality images and the description of every dataset have been rewritten.

Point 5: In the related work section, a more rigorous investigation on the existing methods, such as comparison of previous approaches in terms of pros and cons, should be given. A summary table can be used in this regard.

Response 5: Thank you for your suggestion! We have added a summary table (Table 9) to state the advantages and disadvantages of the methods.

Point 6: The Authors need to present and discuss several solid future research directions.

Response 6: Thank you again for your reminding! We have added the future research directions at the last paragraph of conclusion.

Round 2

Reviewer 1 Report

Thank you for the time and effort which have improved the quality of your manuscript significantly. Please check the following figures before proceeding to the next step. Fig 6. Captions b and c are cropped. Fig 13. Misaligned subfigure c. Fig 17. Different size and border color in f. Because your paper contains plenty of figures, paying a little more attention to details will make it a very well-written paper. I wish you all the best in your future research and well-being. Thank you.

Reviewer 2 Report

The Authors have addressed all the comments.

This manuscript is a resubmission of an earlier submission. The following is a list of the peer review reports and author responses from that submission.

Round 1

Reviewer 1 Report

This paper presents a new algorithm for shadow feature extraction. The paper results interesting; however, it is necessary to clarify the contribution of the proposal. The following comments could help to the authors to improve their work:

It is necessary to analyze in the introduction section the differences with similar algorithms proposed in the state-of-the-art. Specifically with algorithms related to shadow feature extraction. The absence of bibliography related to shadow feature extraction make impossible to observe the real contribution of the proposed algorithm. The proposed algorithm is not specified in detail. It is necessary to show the differences with similar state-of-the-art algorithms. The reported in the paper, looks like a case study, which is corroborated with different metrics. However, it is necessary to compare the results with other similar techniques (or algorithms). e.g., k-means, ML, ISODATA, mixed techniques, probabilistic/ non-probabilistic techniques, or mixed techniques; currently exist many proposals in the open literature. The article does not allow measuring how good the authors' proposal is. Some parts of the paper are confused, e.g., in figure 1 we observe a skateboard, a motorcycle; it looks like, pictures obtained from a personal camera (or other high-resolution device). It is necessary to specify the source of the used images and datasets (attaching the link reference). Today, salt and pepper noise, maybe is not a challenge. This is because we can obtain high resolution images, were this type of noise is not significant. However, it can be a perception of mine, so it is necessary to define the challenges of feature extraction in the article. It is necessary to define the structure of the article, focusing on the Shadow Feature Extraction challenges and the proposed algorithm. It is not easy to identify the specific contributions of the work presented (advantages, disadvantages, etc.).

Reviewer 2 Report

The authors defined the intensity of shadow according to illumination intensity, and proposed an object-based shadow index for remote sensing imagery. 

I would like to comment the following:

1. The symbols and equations are not well organized and written. Math symbols and formulas are not well organized and difficult to understand. For example, Equation (3) seems to write down Equation (2) in detail. Equations (6)-(8) should also be rewritten in the order of description again. Equation (6) actually contains two expressions, and Equation (7) is confusing. 

2. Please check the math symbols carefully. The authors used different symbols on the same things. For examples, $L_d$ vs. $I_d$, $L_e$ vs. $I_e$, $r_i$ vs. $r$, etc. Also, the description in L122 has the error in the symbol. In equation (9), the index b is not found in the body of the equation. 

3. The flow chart in figure 5 should be re-drawn. 

4. The label of the axis in the graphs in figure 1 should be presented. 

5. Please use a unified term. For examples, object-based shadow vs. object-based shadow index, intensity vs. value.

6. Please review the current English sentences carefully. expect vs. except, the ratio of A and B?

7. The sub-caption of figures have not been adequately described. Please remove all the verbs in the sub-caption.

8. For better understanding of the indexes, please present a table comparing all the indexes. 

9. Please present how the authors objectively evaluate the performance of the five indexes in Table 3. The performance should be from the mean opinion score (MOS) or similar ones?

10. The conclusion section should be re-written.